# High resolution DLP stereolithography to fabricate biocompatible hydroxyapatite structures that support osteogenesis

**Jessica S. Martinez** *, Sara Peterson, Cathleen A. Hoel, Daniel J. Erno, Tony Murray, Linda Boyd, Jae-Hyuk Her, Nathan Mclean, Robert Davis, Fiona Ginty, Steven J. Duclos, Brian M. Davis, Gautam Parthasarathy

GE Research, Niskayuna, New York, United States of America

* jessica.s.martinez@ge.com

**Data Availability Statement:** All relevant data are within the article and its Supporting Information files.

## Abstract

Lithography based additive manufacturing techniques, specifically digital light processing (DLP), are considered innovative manufacturing techniques for orthopaedic implants because of their potential for construction of complex geometries using polymers, metals, and ceramics. Hydroxyapatite (HA) coupons, printed using DLP, were evaluated for biological performance in supporting viability, proliferation, and osteogenic differentiation of the human cell line U2OS and human mesenchymal stem cells (MSCs) up to 35 days in culture to determine feasibility for future use in development of complex scaffold geometries. Contact angle, profilometry, and scanning electron microscopy (SEM) measurements showed the HA coupons to be hydrophilic, porous, and having micro size surface roughness, all within favourable cell culture ranges. The study found no impact of leachable and extractables form the DLP printing process. Cells seeded on coupons exhibited morphologies comparable to conventional tissue culture polystyrene plates. Cell proliferation rates, as determined by direct cell count and the RealTime-Glo™ MT Cell Viability Assay, were similar on HA coupons and standard tissue culture polystyrene plates). Osteogenic differentiation of human MSCs on HA coupons was confirmed using alkaline phosphatase, Alizarin Red S and von Kossa staining. The morphology of MSCs cultured in osteogenic medium for 14 to 35 days was similar on HA coupons and tissue culture polystyrene plates, with osteogenic (geometric, cuboidal morphology with dark nodules) and adipogenic (lipid vesicles and deposits) features. We conclude that the DLP process and LithaBone HA400 slurry are biocompatible and are suitable for osteogenic applications. Coupons served as an effective evaluation design in the characterization and visualization of cell responses on DLP printed HA material. Results support the feasibility of future technical development for 3D printing of sophisticated scaffold designs, which can be constructed to meet the mechanical, chemical, and porosity requirements of an artificial bone scaffold.

**Funding:** This work was funded by a research grant from GE Research. The authors Jessica S. Martinez, Sara Peterson, Cathleen A. Hoel, Daniel J. Erno, Tony Murray, Linda Boyd, Jae-Hyuk Her, Nathan Mclean, Robert Davis, Fiona Ginty, Steven J. Duclos, Brian M. Davis, and Gautam Parthasarathy are employed by General Electric (GE), with specific affiliation at GE Research. All authors contributed to the design of the studies and Jessica S. Martinez, Sara Peterson, Cathleen A. Hoel, Daniel J. Erno, Steven J. Duclos, Brian M. Davis, and Gautam Parthasarathy co-wrote the manuscript. The funders had no role in study design, data collection and analysis, decision to publish, or preparation of the manuscript.

**Competing interests:** I have read the journal's policy and the authors of this manuscript have the following competing interests: all authors received research support from GE Research and are employees. This does not alter the authors' adherence to PLOS ONE policies on sharing data and materials. All authors declare no competing interests.

# Introduction

Procedures to repair spine and musculoskeletal injuries are hampered by revision rates of ~35% and continue to be a critical issue in health care [1,2]. Bone is the second most transplanted tissue after blood and ~50% of bone transplantation procedures involve the spine [3,4]. Current medical procedures for bone injuries still utilize auto/allografting techniques [5,6] or synthetic implants, primarily metals (titanium) [7]. Despite autologous bone grafting being named the gold standard of care for bone defect management, it suffers from various problems including donor site morbidity and limited quantity of available bone [8,9]. Allografting, considered the next best clinical option, comes with its own list of downsides too, mainly transmission of infection from the donor to the recipient, high rates of rejection by the receiving tissue, and reduced osteoinductivity [9,10]. For large repairs, such as in spinal reconstruction or hip replacements, titanium implants are used over autologous bone due to the complexity and magnitude of material needed [11] and for its particular ability for meeting load-bearing mechanical needs [12]. Advances in selective laser melting (SLM) techniques have led to the creation of porous titanium implants, allowing for increased cellular integration and biocompatibility with the implant (while maintaining the favorable mechanical properties of titanium) previously noted only on the external surface of titanium implants, at the metal tissue layer interface [13–15]. Despite the use of porous titanium, metal-based implants still do not support full osseointegration and are non-resorbable by the surrounding tissue [16–18]. Titanium implants have also been shown, in some cases, to trigger an immune response (particularly in dental implants), [17,19] and concern over heavy metal toxicity [20] have also been raised. Taken together, the list of issues with auto/allografting and the lack of resorption of titanium implants highlight a need for new scaffold material systems that can stimulate new bone growth at the injury site, ultimately becoming fully replaced by healthy bone.

Bone is a composite material that consists of ~60% hydroxyapatite, 20% collagen, 9% water and several materials and trace elements [21,22]. Bone material is both heterogeneous and anisotropic, classified by two major subtypes based on structural and mechanical properties: trabecular bone and compact bone [23]. A variety of biomaterial approaches have been examined in the literature to fabricate scaffolds that meet the chemical composition and physical architecture of bone [24]. Common techniques include electrospinning, gas-foaming and 3D printing, each with their own advantages and disadvantages and material processing capabilities [25]. While electrospinning has been widely used for polymeric bone scaffolds due to controllable fiber assembly, the technique performs poorly with regard to lack of pore-structure repeatability and ability to produce mechanically strong substrates [26–28]. 3D printing offers the greatest design and overall application flexibility, accommodating a variety of materials (polymers, ceramics, metals, blends), while finely tuning print features and controlling mechanical properties using CAD [27]. Significant work has gone into identifying and characterizing the ideal material system for bone regeneration. For example, synthetic (i.e. poly(lactic acid) (PLA)) and natural (i.e. hyaluronic acid (HA)) polymers [29], proteins [27,28,30], and inorganic materials (i.e. hydroxyapatite) [31] have all been widely characterized for bone regeneration applications. However, to date, none of these described combined material systems and manufacturing approaches fully mimic bone architecture in complexity or in mineral composition [30,32].

Calcium phosphate (CaP) scaffolds are widely studied and known to have suitable osteoconductivity [13], but ceramic materials are brittle and fracture easily. Within the last ten years several commercial efforts have invested in the additive printing of bone implants (i.e. Xilloc, 3DCeram-Sinto, Lithoz GmbH), but they focused on non-load bearing locations, such as the cranium and jaw. The maturation of vat polymerization printing techniques, including DLP,

has enabled fabrication of sophisticated scaffold designs needed to meet the porosity require-ments of an artificial bone scaffold while demonstrating improved compressive strengths over those of traditionally fabricated scaffolds [33]. Coupons were used in this study as an evaluative design to characterize and easily visualize the cellular response to the DLP printed HA mate-rial. The simple geometry facilitated histological assessments (Alizarin Red S and von Kossa staining), fluorescent microscopic analysis, and characterization of cell replication and bio-markers (RealTime-Glo[TM] MT Cell Viability Assay and alkaline phosphatase). In this, paper, we confirm that the DLP process can fabricate biocompatible hydroxyapatite structures that support osteogenesis. These results provide a foundation for engineering complex 3D scaffolds with improved properties for osseointegration of scaffolds. Thus, we propose that DLP-based additive manufacturing techniques could be leveraged to construct ceramic scaffolds contain-ing active physical and chemical components that guide cell differentiation for establishing mature bone tissue.

## Materials and methods

### DLP 3D print sample design

Using the DLP printing system (outlined in Fig 1), a modified circular coupon geometry (17.5mm length, 21mm diameter, and 0.5mm thickness) (see Fig 2A for image of coupon after printing and S1 Fig for illustrations of coupon dimensions and print direction) was selected to maximize the coverage area of a single cell in a standard 12 well plate and provide feasibility for material characterization and biocompatibility. Disk/coupon structures are commonly used in material characterization efforts for potential orthopedic materials. This design geome-try provides an effective method for evaluation of material biocompatibility compared to more

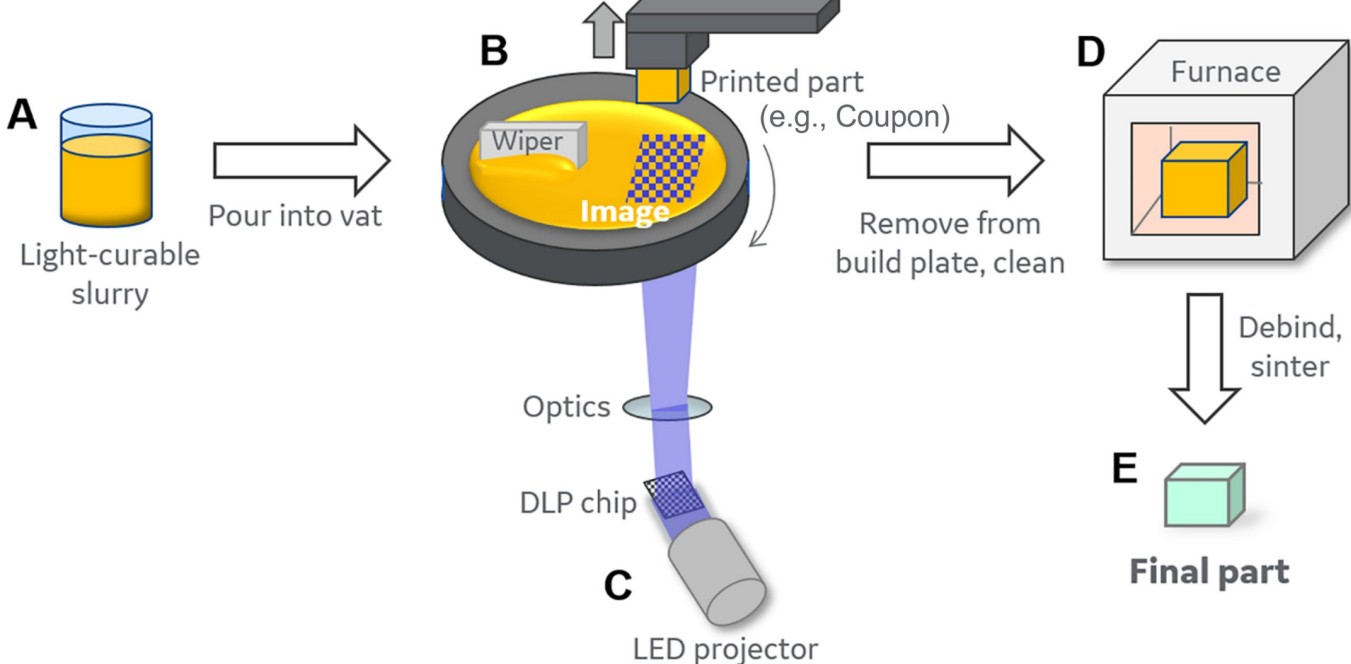

**Fig 1. Illustration of DLP process.** (A) The DLP process starts with a light curable resin and a powder, which are combined to make a photocurable slurry. We utilized a premixed hydroxyapatite photocurable slurry. (B) Slurry is poured into the vat where the (C) light is projected from below onto the build plate, curing one layer at a time of the part (e.g., coupons). (D) The printed part (coupon) is removed from the build plate and cleaned, then placed in a furnace where the light curable resin is burned out and the ceramic densifies, (E) creating the final part (coupon).

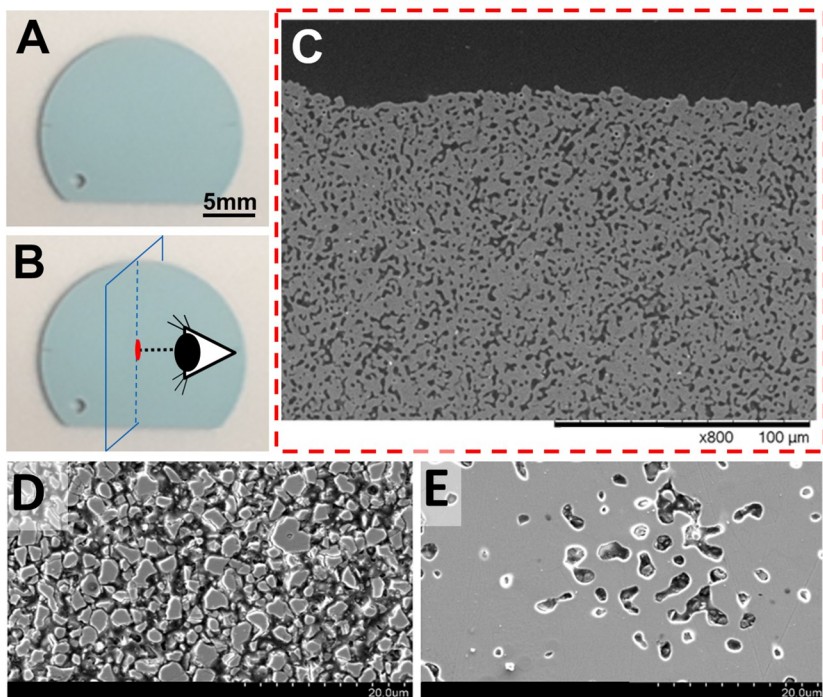

**Fig 2. Scanning Electron Microscopy (SEM) of DLP print builds.** (A) Macro view of coupons built using DLP process, which were rinsed with a propylene carbonate/tripropylene glycol blend and then fired. Micro view and analysis of coupons using SEM imaging was conducted on coupons (B) transversely sectioned mounted and polished, (C) revealing innate porosity. (D) SEM image of coupon before firing (green state, not sintered) reveals particle size of LithaBone HA400 ~2–6 microns and (E) after firing (heat treatment at 1300˚C),densification of particles is seen as a result of sintering.

complex structures. A small section of the circle was truncated to provide a flat edge to begin the print. The coupons went through several design iterations to improve ease of handling, including the incorporation of a small hole in the corner to facilitate ease of transfer using forceps.

## DLP vat polymerization sample manufacturing

Commercial LithaBone HA400 (Lithoz GmbH) slurry, consisting of hydroxyapatite dispersed in a photocurable resin (similar to slurries used in Schmidleithner, C. et al [33], hydroxyapatite (HA) power was suspended in an organic matrix consisting of acrylates and methacrylates with a radical photoinitiator absorbing in the blue visible region [34–36]), was used with the CeraFab 7500 (Lithoz GmbH) where selective exposure to visible light cures the photocurable slurry. According to Lithoz's technical data, the LithaBone HA400 slurry has a solid loading (vol%) of 46 and a dynamic viscosity (Pa-s at 50 s$^{-1}$)of 5–10 and sintered ceramic from these slurries are expected to have a Theoretical density (g/cm$^3$) of 3.16, a Relative density (%) of 85, and final fired composition of hydroxyapatite (HA).This work utilized a DLP printing system where the build plate is above a vat that holds the ceramic slurry (an illustration of the DLP 3D printing process is outline in Fig 1). During the build process, the build plate is lowered into the ceramic slurry and a light image is projected from below the vat, curing the layer. The build plate lifts with the cured layer, the vat rotates to recoat with fresh slurry, and then the build plate is lowered into the slurry to cure the next slice. Nominal dimensions of coupons were 17.5 mm tall, 21 mm diameter, and 0.5 mm thick, see S1 Fig. The coupons were printed

with a 1.13 isotropic scale factor applied to account for firing shrinkage. The coupons were printed such that the 17.5 mm height was built up by the build layers. Print layer thickness of 25 microns was selected and the cure time per layer was 3.5 seconds at full LED intensity (56 mW/cm$^2$). The printed parts were removed from the build plate and cleaned using a blend of 55% propylene carbonate (Thermo Fisher Scientific, Cat No. A15552) and 45% tripropylene glycol (Thermo Fisher Scientific, Cat No. 31798) (v/v%). After cleaning, prints were fired in two steps. The first step was a low temperature firing to 205˚C using a DKN602C oven (Yamato) to remove volatile organic components prior to high temperature sintering. The second step was a high temperature firing to sinter the parts to 1300˚C using a L7S Rapid Temp tube furnace (CM Furnaces). For this study, all coupons were fired in an alumina tube furnace with flowing $N_2/O_2$ (80/20 volume mixture). The coupons were light blue in color, consistent with vendor data sheet, see Fig 2A. All firing steps (time, temperature) followed the Lithoz GmbH provided manuals for processing HA400 material.

## Phase identification, density, and characterization

Powder X-ray Diffraction (XRD) and Fourier-transform Infrared Spectroscopy (FTIR) was used to identify the crystalline phases of the fired coupons. Fired coupons were crushed into a powder and analyzed by a Bruker D8 Discover machine for XRD and with a Bruker Optics, Hyperion 3000 IR microscope attached to a Bruker Optics, Vertex 70 FTIR. For FTIR, the powder was pressed between diamond plates to obtain a thin film for the transmission measurement in the Hyperion 3000 IR microscope. The NIST standard spectrum for calcium phosphate hydroxide was used for comparative analysis (NIST website: https://webbook.nist.gov). The X-ray tube used was the IμS Microfocus Source using Cu K radiation (50 kV and 1mA). HELIOS multilayer optics and a 0.5 mm collimator were used to focus the incidence beams. The sample was mounted on the universal sample stage, set in the reflection geometry, and oscillated along the X and Y directions. The data was collected with the VANTEC-500 2D area detector for 30 minutes for each coupon. Inductively Coupled Plasma-Optical Emission Spectroscopy (ICP-OES) analysis was conducted on LithaBone HA400 powder. Samples were prepared for ICP-OES analysis by hot block digestion. All data was collected using a spectro ICP-OES instrument with a CETAC Autosampler ASX-520, Tracey Teflon cyclonic spray chamber, MiraMist nebulizer and D-Torch with ceramic outer tube and alumina inner tube, tapered alumina injector. Sample preparation was as follows: LithaBone HA400 powder was ground with a mortar and pestle and 0.05 g of powder was added to a pre-weighed 50 mL natural cap tube. To this, 4 mL of $HNO_3$/HCl 25/75 vol% was added to the tube and the tubes were placed inside a heat block at 90˚C for 30 minutes to allow for digestion. After incubation, deionized water was added to the 50 mL fill line on the tubes and the tubes were re-weighed. A 100-fold dilution was made of the prepared digestion solution using 3% HCl, 1% $HNO_3$, 0.5% HF in 15ml natural cap tubes. Internal standard of Sc was added to matrix (1ml), followed by acid and filled with deionized water to the 10ml gradation mark. ICP-OES analysis results for Ca, P, and Ca/P ratios are shown in S2 Fig.

Bulk density measurements were completed on HA coupons by Archimedes principle. Specimens were weighed first in air on a precision micro-scale balance to measure the dry weight (D). Samples were weighed in degassed water to measure the immersed weight (W). Samples were removed from water and dabbed lightly with a wet wipe to remove excess liquid and weighed to determine the saturated weight (S). The bulk density ($\rho_B$) and open porosity (P) of the sintered coupon was calculated based on Eqs 1 and 2, respectively, where $\rho_w$ is the

density of water.

$$\rho_B = \frac{D}{S - W} * \rho_w \qquad \text{Eq 1}$$

$$P = \frac{S - D}{S - W} \qquad \text{Eq 2}$$

Surface roughness of the HA coupons was measured by white light profilometry using a Keyence VR-3000 profilometer. The center of each coupon was measured for multiline roughness over 5 parallel lines in both the vertical and horizontal direction along a 2 mm line length to obtain an average roughness value $R_a$. Samples were processed and roughness analyzed using the standard Keyence software for the profilometer, see S3 Fig. SEM images were collected under vacuum in electron backscattering mode on a Hitachi TM3030Plus benchtop electron microscope, see Fig 2C. Contact angle was measured by dropping 2 μL of deionized sterile water onto the coupons and measuring the angle using VCA Optima instrument (AST Products Inc). The optical image was manually taken 2 seconds after the water drop encountered the coupon and both the right and left angles of the water droplet were measured using computer software. Three measurements were taken per coupon. Coupons were characterized as hydrophilic (contact angle less than 90 degrees) or hydrophobic (contact angle greater than 90 degrees), see S4 Fig.

Flexural strength measurements of sintered DLP 3D printed LithaBone HA400 modulus of rupture (MOR) test bars were conducted using a four-point bend flexural set-up. As depicted in S10 Fig, MOR bars (L x w x d: 40 mm x 3 mm x 0.5 mm as-printed) were printed in both the vertical and horizontal direction. An example build plate setup for printing test bars is shown in S10A Fig. Bars were printed on the same build plate, cleaned in the same solvent bath (blend of 55% propylene carbonate and 45% tripropylene glycol), and fired in the same furnace to isolate the effect of print orientation on sintered LithaBone HA400 fired strength. It's important to note that dimensions of MOR test bars used differ from ASTM testing standard C1161 because thicker bars printed from LithaBone HA400 crack during firing. To avoid this, the width of our test bars was increased, providing more area under the applied load to compensate for the thin depth. The four-point bend flexural testing set-up used in these efforts is illustrated in S10A Fig.

Flexural strength was measured using a four-point bend test fixture with articulating rollers per ASTM C1161. The load frame was an Instron 4465 series with electro-mechanical 2 ball-screw frame and was operated using Instron WaveMatrix Series IX Software. Instrumentation set up was as follows: load span of 10 mm, support span of 20 mm, and displacement ramp rate of 0.01 in/min. Compression was applied via the rollers until the sample fractured and the maximum flexural strength was calculated using the equation below where F is the force in Newtons at cracking, L is the support span, w is the width of the test bar in mm, and d is the depth of the test bar in mm.

$$\text{Max Flexural Strength (MPa)} = \frac{0.75 \cdot F \cdot L}{w \cdot d^2}$$

DLP scaffold designs (clover and trifurcating) from LithaBone HA400, proposed for bone formation studies (see S10C Fig) were imaged using a Hirox 3D digital microscope. Scaffolds seeded with U2OS cells (stained with nuclear fast, used in von Kossa staining protocol) were also imaged.

## Preparation of coupons for biological studies

Individual coupons were placed into an autoclavable/gamma/EtO sleeve (Fisher Scientific, Cat No. 0181250) prior to autoclaving at 121°C for 30 minutes with steam or 25kGy gamma irradiation (Steris). To confirm sterility, coupons were placed in tryptic soy broth (Teknova, Cat No. T1515) for 14 days at 25°C with rotation set to 120 rpms in a Max Q 5000 (Thermo Fisher Scientific). Tubes were checked/photographed daily for turbidity and on day 14, final day of incubation, liquid broth absorbance was measured at 600nm using cell culture parameters on Nanodrop 2000 (Thermo Fisher Scientific). To enhance initial cell attachment, coupons were primed with fibronectin at $30\mu m/cm^2$ (Sigma Aldrich, Cat No. F4759-1MG) or Matrigel 800μl/well (12-well plate) of a 2.5% v/v% solution of Matrigel in 1x PBS (Corning Inc. Cat No. 356234).

Extractables and leachables (E & L) were evaluated by coupon immersion in standard culture media for 3 days under standard incubation conditions (5% $CO_2$ at 37°C). In a 12-well tissue culture plastic format (Corning) U2OS cells were seeded at 60% confluency ($4.2x10^3$ cells/ $cm^2$) 24 hours before addition of "coupon conditioned media" for 48 hours. Phase contrast microscopic analysis was used to evaluate cell viability and morphologic indicators of cell health including size, shape and lack of apoptotic bodies.

## Mammalian cell culture

Two adherent cell lines were used for *in vitro* studies: U2OS (ATCC, HTC-96) a human osteosarcoma cell line derived from bone with an epithelial morphology and human mesenchymal stem cells (MSC) (Lonza, Cat No. PT2501). U2OS cells were grown at 37°C with 5% $CO_2$ in McCoy's' 5a (Modified) Medium with HEPES (Life Technologies Corporation, Cat No. 12330031) supplemented with 10% Hyclone Fetal Bovine Serum (General Electric Healthcare Bio-Science Corp, Cat No. SH30088.03), 1% Antibiotic-Antimycotic (100x) (Life Technologies Corporation, Cat No. 15240062), and 0.1% of Gentamicin (10mg/mL) (Life Technologies Corporation, Cat No. 15710064). U2OS cells were subcultured 2–3 times per week at a ratio of 1:6 using trypsin-EDTA (1x) solution (ATCC Cat No. 30–2101). MSCs were cultured at 37°C with 5% $CO_2$ in MSCGM™ Mesenchymal Stem Cell Growth Medium with supplements (Lonza, Cat No. PT3001). Subculture of MSCs was done using the ReagentPack™ Subculture Reagents (Lonza, Cat No. CC-5034).

## Osteogenic differentiation

MSCs underwent osteogenic differentiation in Human Mesenchymal Stem Cell (MSC) Osteogenic Differentiation Media (Lonza, Cat No. PT-3002). Cells were seeded at $2x10^4$ cells/$cm^2$ in a 12-well tissue culture plate (Corning, Cat No. 3513; well diameter 22.1mm) and cultured in osteogenic differentiation media for 14–35 days. On day 14, cells were assayed for osteogenic differentiation using alkaline phosphatase assay kit (Abcam, Cat No. ab83369) and Alizarin Red S staining (Abcam, Cat No. ab146374). On day 21, mRNA was extracted from the cells using Direct-zol RNA Miniprep kits (Zymo Research, USA, cat no. R2050) and analyzed for expression of osteogenic differentiation makers Runx2, Sp7 (osterix), and Spp1(osteopontin) in comparison to GAPDH (control). RT-PCR primer targets used the following Taqman probes: GAPDH (Hs02758991_g1), Sp7 (HS00541729_m1), Spp1 (Hs00959010_m1), and Runx2 (Hs010479769_m1). RT-PCR was performed using TaqMan™ Fast Advanced Master Mix (Thermo Fisher Scientific, USA, cat no. 4444556) on QuantStudio 6 and 7 Flex Real-Time PRC systems (ThermoFisher). On day 35, cells were stained with von Kossa stain (Abcam, Cat No. ab150687).

## Culturing cells on DLP printed coupons

Sterilized coupons were placed into 12-well tissue culture plastic plates (Corning, Cat No. 3513) and washed three times with 1mL of 1x PBS buffer (Thermo Fisher Scientific, Cat No. 10010023). Cells were immediately seeded onto the coupon surfaces and were adhered overnight, then the coupons were transferred into a new well. Coupon transfer was done to ensure cells that adhered to the underlying tissue culture plastic surface did not impact results. Seeding efficiency was estimated by quantification of unattached cells. Proliferation of cells on coupons was evaluated using RealTime-Glo™ MT Cell Viability Assay (Promega, Cat No.G9711). Cell attachment morphology was evaluated by microscopic analysis using phase contrast imaging (EVOS, Thermo Fisher), low magnification bright field (Leica dissection microscope), and immunohistochemistry (IN CELL analyzer, GE Healthcare). For immunohistochemistry, cells were processed for staining using Image-iT™ Fixation/Permeabilization Kit (Thermo Fisher Scientific, Cat No. R37602) and stained with Alexa Fluor® 647 conjugated Ribosomal Protein S6 Antibody (Santa Cruz Biotechnology Inc., Cat No. sc-74459 AF647), Phalloidin CruzFluor™ 488 Conjugate (Santa Cruz Biotechnology Inc., Cat No. sc-363791), and ProLong™ Gold Anti-fade Mount with DAPI (Life Technologies Corporation, Cat No. P36935). Post image analysis was done on ImageJ (image processing software, can be downloaded from https://imagej.nih.gov/ij/).

## Results

### Simple geometrical 3D DLP printed, coupon design

Coupons in this paper were 3D printed using the DLP manufacturing process, illustration shown in Fig 1, utilized a commercial LithaBone HA400 (Lithoz GmbH) slurry. Coupons were printed with simple geometry, designed for effective handling in single wells of tissue culture polystyrene (TCP) plates (12-well) and evaluation of the coupon material for the potential use of the DLP process in future development of more complex geometries. LithaBone HA400 slurry poured into a vat and light, projected from below and onto the build plate, cures, one layer at a time, each layer 25μm thick. The coupon prints were built to have 17.5mm length, 21mm diameter, 0.5mm thickness. Coupons, circular in shape, were also designed with a hole on the edge to facilitate handling with sterile forceps (S1A Fig), which facilitated transfer of coupons in and out of single wells within TCP plates. Coupon geometry was also designed with a flat edge, where coupon print is adhered to the build plate and the layers grow outward in the direction of the arrow shown in S1B Fig. This geometrical design proved effective for minimizing print failures and damage during removal from build plates, all while maximizing the available space for printing coupons on a single build plate. Printed coupons were cleaned, then heated to remove the light curable resin and sinter the ceramic. The simple coupon design presented in this paper, allowed for ease of material characterization and evaluation of biocompatibility of hydroxyapatite DLP printed material. Use of simple coupon designs are commonly employed for evaluation of potential orthopaedic materials.

### Phase identification, density, characterization, and contact angle measurements

Fired coupons were analyzed using powder X-ray diffraction (XRD). The XRD pattern was consistent with hydroxyapatite (HA), demonstrating coupons to be predominately HA with very minor secondary phases as shown in S2 Fig. The IR spectrum obtained by FTIR shows expected peaks for hydroxide (OH⁻) and phosphate $(PO_4)^{-3}$ as compared to the NIST standard spectrum for calcium phosphate hydroxide. ICP-OES was also conducted, and measured Ca/P

ratios were on average 1.66, see S2C Fig. The bulk density averaged across 10 coupons was 2.71 $g/cm^3$ with a standard deviation of 0.13 $g/cm^3$. The average open porosity of the same 10 coupons was 4.4% with a standard deviation of 3.5%. Measured open porosity values across the coupons ranged from 0.0% to 9.6%, showing a relatively broad range of final open porosity values. The variation in open porosity was expected and potentially resulting from thermal gradients in the tube furnace and the material's sensitivity to the maximum sintering temperature. Controlling thermal gradients during sintering may improve the uniform distribution of material porosity and slurries can be engineered to produce sintered prints with target porosities [37]. In addition, a four-point bend flexural test study was conducted, to probe at the inherent strength of LithaBone HA400 prints. Using MOR test bars with dimensions that differ from ASTM testing standard C1161 (see S10 Fig), the printing direction had a direct impact on flexural strength. Not surprisingly, the inherent material strength of LithaBone HA400 (~63 ±6MPa, N = 10, horizontal flexural strength and ~16±7MPa, N = 7, vertical strength) was lower than reported ranges for cortical bone (100-150MPa) [38]. However, a more in-depth investigation assessing impact of LithaBone HA400 print geometry on its strength, particularly for scaffolds whose structure has been optimized for load-bearing applications, would be most informative regarding the potential use of LithaBone HA 400 in bone formation.

Light profilometry was used to measure surface roughness of DLP printed coupons. An example roughness profile of an HA coupon is shown in S3 Fig for both parallel (vertical) and perpendicular (horizontal) directions with respect to the build direction. Measured across 10 coupons, the average vertical Ra value was 0.40 μm with a standard deviation of 0.10 μm and an average horizontal value of 1.09 μm and a standard deviation of 0.15 μm. Coupons appear to have a rougher surface perpendicular to the build direction, and a smoother surface parallel, indicated by the higher average Ra value in the horizontal direction and lower values vertical to the build direction and may be an artifact of the DLP 3D printing process. The roughness of the coupons is far less than that of the build layer thickness (25μm) and closer to that of the primary particle size.

Scanning Electron Microscopy (SEM) images were also taken of coupons to analyze internal porosity and surface roughness. Coupons built using the DLP process were transversely sectioned, mounted, polished, and analyzed by SEM to reveal pore size and distribution of porous HA coupons, see Fig 2. SEM image of coupon also shows densification of LithaBone HA400 particles (measured to be ~2–6 micron in green state—not sintered prints) after firing (heat treatment at 1300˚C), see Fig 2C and 2D. DLP printed HA coupons were measured for contact angle (S4 Fig) and were determined to be hydrophilic, with an average contact angle of 69˚ and a standard deviation of 20˚ when calculated across 10 coupons, averaging the left and right angles into a single value.

## Sterility, leachables and extractables from printed coupons

The DLP manufacturing process was not conducted in a sterile environment, therefore post-manufacturing sterilization was required. Sterilization using 25kGy gamma irradiation and standard autoclave methods were evaluated. To assess bacterial contamination, coupons were maintained in tryptic soy broth for 14 days at 25˚C with shaking. Tubes were checked daily for turbidity and on day 14 the liquid broth absorbance was measured. None of the tested coupons had visible turbidity or an increase in absorbance, indicating no microbial growth from the sterilized coupons using either sterilization method.

To address potential leachable and extractable production during the DLP additive manufacturing process (e.g. impurities in the slurry, residual organic material from the printing process, byproducts of sintering, or inorganic contaminates), autoclaved coupons

were exposed to cell culture medium for 3 days. U2OS human osteosarcoma cells, a routinely used and well characterized cell type that mimics mesenchymal stem cells in size and attachment and growth characteristics, were cultured in coupon conditioned medium for 48 hours. The cells maintained a normal morphology and there was no increase in cell detachment from the plates nor acquisition of apoptotic bodies within the cells (S5 Fig). Thus, we did not observe any evidence for extractables and leachables generated that overtly altered cell morphology and phenotype.

### Cell seeding and morphology on coupons

U2OS cell attachment efficiency on coupons was comparable to standard tissue culture polystyrene plates. Seeding $3.5x10^3$ cells/cm$^2$ cells onto HA coupons (diameter 21mm) and 12-well plate (well diameter 22.1mm) both resulted in approximately 50% confluency after overnight culture. Measurements of unattached cells in the supernatant were found to be negligible for both conditions. Post-seeding, HA coupons were transferred to a different well within the tissue culture plate. Cells did not migrate off the HA coupon and onto the underlying supportive polystyrene surface, demonstrating that the HA coupon was a favorable surface for cell culture. The morphology of U2OS on coupons and standard culture plates were comparable.

In contrast, attachment efficiency of MSCs on HA coupons was reduced by 30% when compared to standard tissue culture polystyrene (S6A Fig). We speculated that pre-adsorption of extracellular matrix proteins onto the HA coupon would increase MSC attachment using a standard immersion procedure as used in Deligianni, et. al. [39]. The attachment efficiency of MSCs on coupons was greatly improved when coupons were coated with fibronectin or Matrigel (S6 Fig). The morphology of MSCs on HA coupons was evaluated by DAPI, phalloidin and S6 immunofluorescence (Fig 3). MSCs cultured on coupons were observed to have a spindle morphology consistent with healthy undifferentiated MSCs, confirming biological compatibility of the DLP printed HA materials. MSCs cultured on polystyrene obtained a standard fibroblastic morphology. Overall, MSC morphologies were as expected across the two surfaces.

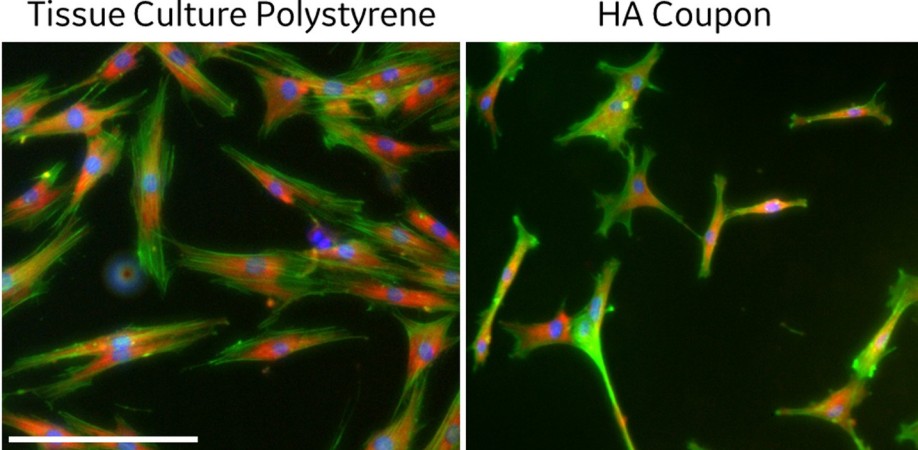

**S6 (Ribosomal Protein); Phalloidin (Actin); DAPI (Nuclei)**

**Fig 3. MSCs morphology on DLP printed coupons.** MSCs seeded onto tissue culture polystyrene (12-well plate) and DLP printed HA coupon were fixed and stained for S6 ribosomal protein (red, conjugated primary antibody), actin (green, phalloidin dye), and nuclei (blue, DAPI) after 24 hours of culture. Image taken at 20x objective magnification using IN CELL analyzer, demonstrates a typical spindle morphology on coupons. Scale bar 50μm.

## Proliferation rate of U2OS and MSCs on coupons

DLP printed HA coupons were opaque, which warranted a non-microscopic quantification approach for assessing cell density and proliferation. Therefore, cell proliferation was quantified using an indirect quantification assay (RealTime-Glo[TM] MT Cell Viability Assay, Promega) in which a substrate is enzymatically converted to a luminescent product via mitochondrial metabolism. This enables continuous non-destructive quantification of cell metabolism over time, providing an indirect estimate of cell density on coupons and cell culture plates over 48–72 hours. A standard curve was generated to correlate relative luminescence units (RLUs) with cell number (S7 Fig). Using this assay, U2OS cells seeded at roughly ~25% ($1.75x10^3$ cells/cm$^2$) confluence in 12-well tissue culture plates and on HA DLP printed coupons were observed to have comparable proliferation rates (P>0.5; using T-test, differences were not statistically significant) (Fig 4). At 66 hours post-seeding, cells were fixed, and a direct count was performed on DAPI stained nuclei, confirming equivalence of cell proliferation on HA coupons.

Similarly, MSC proliferation on HA DLP printed coupons was comparable to polystyrene (Fig 5). MSCs ($1.0x10^4$ cells/cm$^2$) were seeded at approximately 40% confluence onto tissue culture polystyrene 12-well plates and coupons and proliferation over 3 days was quantified using the RealTime-Glo[TM] MT Cell Viability Assay (Promega). We observed that MSCs on uncoated HA coupons had a reduced attachment efficiency compared to coupons precoated with fibronectin or Matrigel (S6 Fig). Nonetheless, cell doubling time was similar for MSCs on HA coupons and standard tissue culture polystyrene plates at all seeding efficiencies (S7 Fig).

## Osteogenic differentiation of MSCs on coupons

MSCs ($2.0x10^4$ cells/cm$^2$) were seeded on coupons or 12 well plates (well diameter 22.1mm) at roughly 80% confluency, expanded for two days to reach confluency, then were maintained for 14 to 35 days in either osteogenic differentiation medium or MSC growth medium.

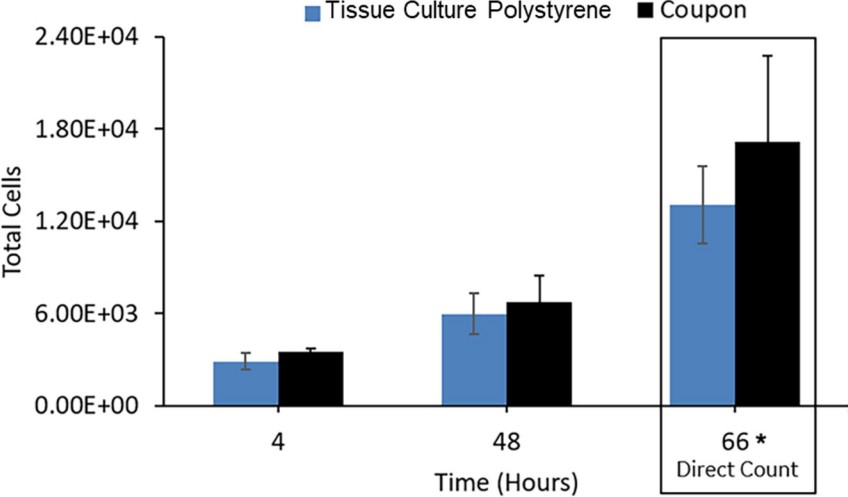

**Fig 4. U2OS proliferation on DLP printed HA coupons.** Relative luminescence unit (RLUs) were measured from U2OS cells seeded at ~25% ($1.75x10^3$ cells/cm$^2$) density on polystyrene plates or DLP printed HA coupons using RealTime-Glo[TM] MT Cell Viability Assay. Cell counts were enumerated based on RLU measurements for the 4 and 48 hour time points using a standard curve correlating cell count and RLU (S7 Fig). Direct cell count was done at 66 hours of incubation by image analysis (counted cell nuclei). U2OS cells proliferation rates were similar on coupons and tissue culture polystyrene plates (P>0.5; using T-test, differences were not statistically significant).

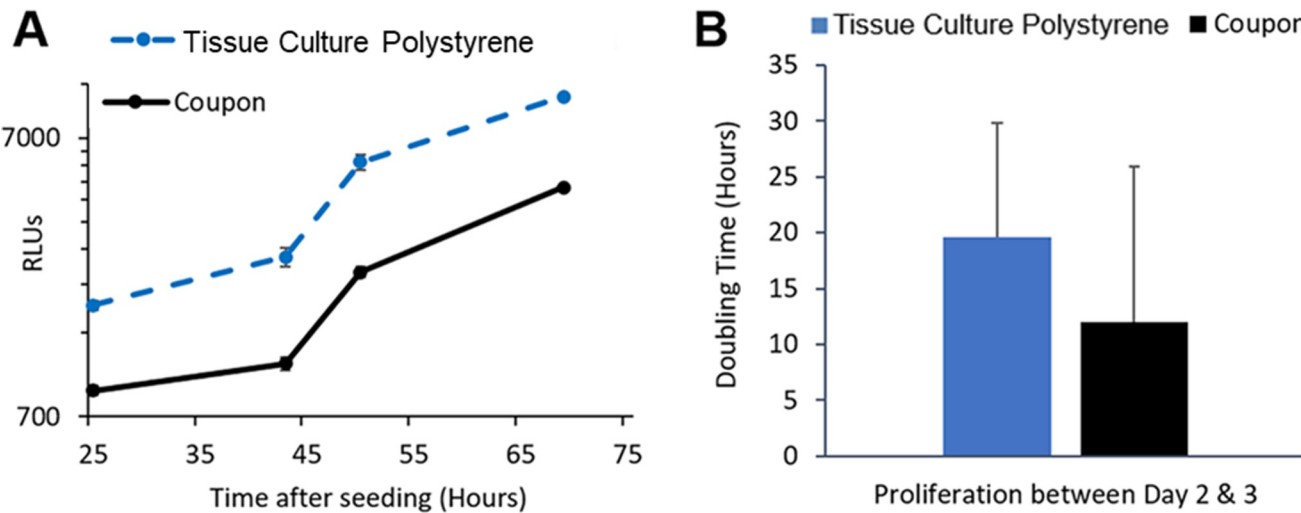

**Fig 5. MSC proliferation on DLP printed coupons.** (A) MSCs were seeded at an approximate density of 40% ($1\times10^4$ cells/cm$^2$) onto tissue culture polystyrene (12-well plates) and coupons. Cells were cultured for 3 days in the presence of RealTime-Glo™ MT Cell Viability Assay (Promega). Relative luminescence unit (RLUs) was measured at various time points and plotted using a logarithmic y-axis. (B) Cell doubling times were comparable to cells attached on tissue culture polystyrene plates.

Osteogenic medium resulted in cells obtaining both osteogenic (geometric, cuboidal morphology with dark nodules) and adipogenic (lipid vesicles and deposits) features (Figs 6 and S8 and S9). Osteogenic differentiation was assessed on day 14 by Alizarin Red S staining for calcium deposits. On day 35, von Kossa staining for calcium deposition was performed. While calcium deposition assays were complicated by the presence of calcium phosphate in HA coupons and the opacity of the coupons, the increased frequency and size of gray/black punctate staining

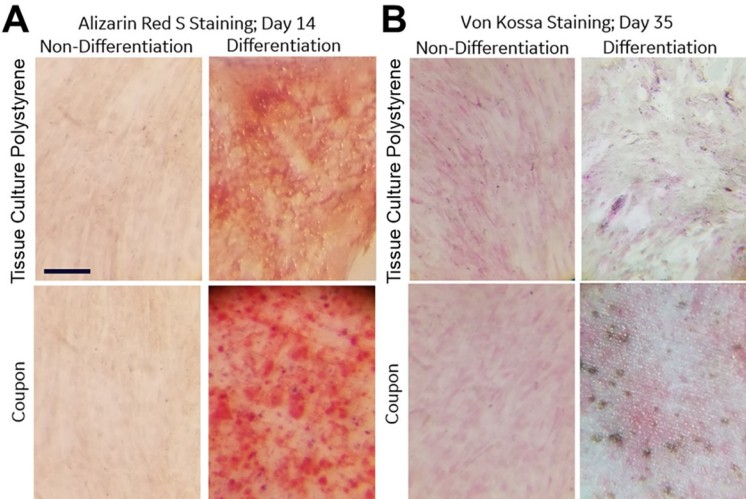

**Fig 6. Calcium deposition from osteogenic differentiated MSCs on DLP printed coupons.** (A) Calcium deposits stain red when cells are stained with Alizarin Red S and (B) gray to black with von Kossa. (B) The von Kossa counter stain, nuclear fast red, stains cells a fuchsia color. The observation of punctate staining in MSCs cultured in osteogenic differentiation medium on DLP printed coupons and polystyrene plates demonstrates calcium deposition on both surfaces using both Alizarin Red S and von Kossa staining. In contrast, MSCs cultured in growth media lack similar punctate calcium staining with either Alizarin Red S or von Kossa. Images were taken using Lecia dissection scope at 4x magnification, scale bar 1.25mm.

for von Kossa and dark red for Alizarin Red S, indicates cellular calcium deposition on DLP printed coupons. MSCs cultured in standard growth media did not show similar punctate calcium staining with either alizarin Red S (S8 Fig) or von Kossa (S9 Fig).

In addition, MSCs cultured in osteogenic medium for 14 days exhibited 3.5–4 folder higher (P<0.05) alkaline phosphatase activity than MSCs in MSC growth medium on both DLP printed HA coupons and standard polystyrene tissue culture plates (Fig 7A). Similar fold increase was observed for mRNA expression of Runx2 as compared to GAPDH (control) (P<0.05), an early stage osteogenic differentiation marker [40] for MSCs cultured in osteogenic medium for 21 days, but not for SP7 and SPP1, both considered late-stage osteogenic differentiation markers [41–44] (Fig 7B). The post-differentiation increase in Runx2 mRNA expression was higher for MSC differentiated on HA coupons as compared with MSC differentiated on standard polystyrene tissue culture plates, while the expression of osterix (SP7), and osteopontin (SPP1), was lower for MSC differentiated on HA coupons. Thus, the material itself may influence the kinetics of differentiation. This pattern of expression resembles an in vivo-like response, where Runx2, an early osteogenic differentiation marker [40,45], peaks near day 21 as compared to SP7 [41,42], and SPP1 [46,47], both later differentiation markers. In all cases, the osteogenic differentiation increased in differentiation culture conditions (Figs 6 and 7A), with increased expression of differentiation markers when compared to GAPDH (control) (Fig 7B) and positively correlated with staining of Alizarin Red S (Figs 6 and 7A and S8) or von Kossa (Figs 6 and S9). These data confirm MSC osteogenic differentiation is supported on DLP printed HA coupons.

MSCs seeded onto DLP printed coupons (two different batches: coupon A and coupon B and average "coupon") or tissue culture polystyrene, "TCP" (12-well plates, TCP) were cultured in non-differentiation media or in osteogenic differentiation media and the (A) fold-increase in alkaline phosphatase activity on day 14 (n = 3) for cells cultured in osteogenic differentiation media was measured and (B) cells were processed for mRNA extraction and analysed by RT-PCR for gene expression of Runx2, SP7, SPP1, and GAPDH on day 21 (n = 2). MSCs were cultured either in osteogenic differentiation media or non-differentiation media. Ratio of differentiation for alkaline phosphatase was calculated by measuring absorbance at OD 405nm. Post-differentiation fold increases in mRNA expression of targeted genes were calculated by measuring RT-PCR ct values of cells cultured in osteogenic differentiation media over that of cells in non-differentiation media. A similar elevation in alkaline phosphatase

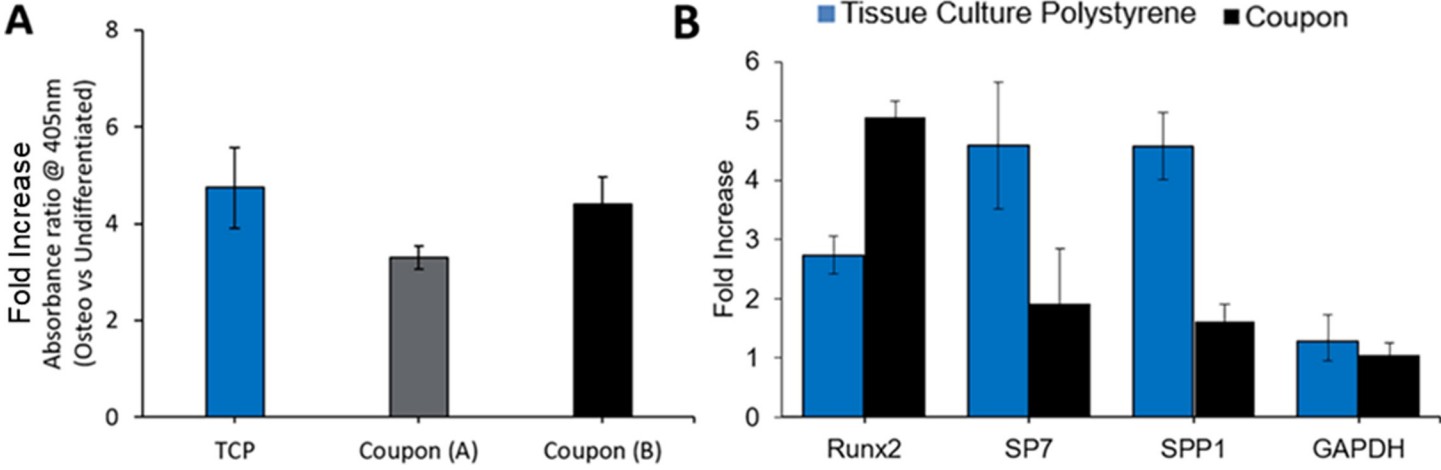

**Fig 7. Expression of alkaline phosphatase and RT-PCR analysis for MSCs differentiated on DLP printed HA coupons.**

activity and Runx2 mRNA expression as compared to GAPDH (control) (P>0.05) was observed for cells on coupons and on tissue culture polystyrene, "TCP", indicating comparable osteogenic differentiation efficiency both early markers for osteogenic differentiation, but not for SP7 and SPP1 mRNA expression, both considered late-stage differentiation markers.

## Discussion and conclusion

High resolution DLP stereolithography holds promise as a versatile technique for the fabrication of bone substitutes. The technique provides an effective platform for the construction of simple geometries, such as our coupon design for material evaluation and analysis of biocompatibility and has shown to be capable of constructing more intricate designs for scaffold development [23,30,33]. The data reported here demonstrates proof-of-concept for 3D-printing of hydroxyapatite coupon structures that support cell seeding, proliferation and osteogenic differentiation. Results using the simple coupon geometry helped effectively evaluate potential use of the DLP process for future development of complex architectural features of HA scaffolds. Notably, as reported in this study and from other recent literature, the DLP process is highly compatible with biological growth [33,48,49]. Utilizing two relevant cell types, the osteosarcoma line U2OS and primary mesenchymal stem cells, we demonstrated cell morphology and cell proliferation rates on the HA coupons that are comparable to tissue culture polystyrene plates. Further we demonstrate evidence for osteogenesis on the HA coupons comparable to other HA based materials [15,49,50].

MSC and U2OS proliferation rates over time were proven to be similar on HA DLP printed coupons and standard tissue culture polystyrene. Cells were seeded onto HA coupons in polystyrene plates, and most seeded cells remained attached to the HA coupons over 35 days, demonstrating preference for HA coupon over polystyrene. During prolonged culture we observed cell migration from the edge of the coupons onto underlying tissue culture plastic, but this can be attributed to empty space in the polystyrene area supporting the coupon. A reduction in lower initial seeding onto the coupons compared to control plates was observed for MSCs but not U2OS cells. We found that priming coupons with fibronectin, or Matrigel (S6 Fig) enhanced initial cell attachment of MSCs to coupons. We propose that all future investigation using HA DLP printed materials would benefit from priming with similar extracellular matrix proteins, a common application used in hydrogels [29] and other engineered materials to improve biocompatibility [8,27,31]. The observed biocompatibility of DLP printed HA coupons supports other investigations using different primary cells [33]. Together, the biological results utilizing DLP printing of commercial LithaBone HA400 confirm the slurry an acceptable material for scaffolds used for osteogenic applications.

We demonstrate that the HA DLP printed coupons supported efficient osteogenic differentiation of MSCs. The morphology of MSCs cultured in osteogenic medium for 14 to 35 days were notably different than MSCs maintained in MSC expansion medium. Osteogenic medium resulted in cells obtaining both osteogenic (geometric, cuboidal morphology with dark nodules) and adipogenic (lipid vesicles and deposits) features. Specific assays for Alizarin Red S staining, von Kossa staining and alkaline phosphatase demonstrate acquisition of osteogenic phenotypes. Interestingly, MSCs in osteogenic differentiation medium reorganized on both polystyrene plates and HA coupons, resulting in cell morphology contraction, but this did not negatively impact osteogenic differentiation (Figs 6, 7 and S8 and S9). RT-PCR analysis of osteogenic differentiation markers Runx2, osterix (Sp7), and osteopontin (Spp1) on day 21 of cultured showed elevated expression of all markers in comparison to control marker GAPDH for MSCs cultured in osteogenic differentiation media. Runx2 is a transcription factor essential for initiation of osteogenic differentiation, with initial high expression that

diminishes over time and peaks between day 14 and day 21 of culture [43,44,51]. Osterix (transcription factor downstream of Runx2) [52,53], and osteopontin (important factor in endocrine-regulated bone formation [54]) are all considered late differentiation makers with their highest levels of expression after weeks of culture [55]. The differences in gene expression on HA coupons versus polystyrene perhaps suggest an influence of the material on the kinetics of the differentiation process, which may be used to control timing to achieve maximum benefit.

After 35 days of culture, in both native tissue culture and HA coupons, the MSCs contracted leaving areas along the surface free of cells with the degree of contraction varying between donor MSC cell lines purchased from Lonza (MSC cell line donor variance were expected [56]), but the level of contraction was not notably different between tissue culture plastic controls and HA. Osteogenic differentiation of MSCs on both tissue culture plastic and HA coupons had a comparable degree of contraction. We found precoating plates and HA coupons with fibronectin or Matrigel eliminated contraction of MSCs and provided an extracellular matrix support for cellular attachment, proliferation and differentiation.

The data presented here confirm the performance of the DLP process with LithaBone HA400 slurry for 3D printing of hydroxyapatite coupons that support cell seeding, proliferation and differentiation. Results from this study are consistent with similar observations reported by Schmidleithner et al. [33] and shows similar biocompatibility with other calcium phosphate materials [49] and engineered scaffolds [24,50]. Future work will leverage these results toward design and cell performance on 3D scaffolds with complex internal structures that mimic trabecular bone; some examples are depicted in S10C Fig, which represent unit cell structures we have successfully printed by DLP that we believe may have relevance for supporting bone regeneration. Within these examples, we can print features with micron control (~40μm resolution with minimum feature size ~150μm for complex parts) and regulate the local topologies experienced by seeded cells. Notably, designs can be tailored to have architectural elements promoting cellular integration and responsiveness [25]. The 3D scaffold structures will be designed to support cell growth, vascularization and bone formation, while pushing towards mechanical properties, including compression and rigidity, comparable to bone. Using a four-point bend flexural study and rupture test bars with dimensions that differ from ASTM testing standard C1161, we were able to measure flexural strength of sintered LithaBone HA400 prints, see S10 Fig. Not surprisingly, the inherent strength of LithaBone HA400 is lower than that of human cortical bone. As shown in S10A Fig, decreased vertical strength was expected due to build layer orientation in the four-point bend test setup. Decreased strength due to build layers sitting parallel to the fracture axis (vertical) as opposed to perpendicular (horizontal) has been shown by anisotropic mechanical behavior of ceramic additive components and is well reported in the field of additive manufacturing [57]. Material defects (e.g. build layer cracking and interlaminar porosity) can disproportionally impact the flexural strength measurement in the vertical direction. Both print orientation and geometry are variables for controlling and optimizing the mechanical properties of additively printed materials [58], including LithaBone HA400 prints [59,60], and should be further investigated. In addition, changes to inherent strength of LithaBone HA400 dependent on scaffold geometry should be analyzed, with an in-depth investigation into its potential usability in both loading-bearing and non-loading bearing orthopedic applications. DLP constructed scaffolds should be evaluated for mechanical properties including its strength, compression, bending and biaxial bending, which will provide relative insight into its potential application for clinical use. In this study, a commercial slurry from Lithoz was used, however, future work should also consider integration of other slurry components (biphasic material systems) to improve the materials osseointegration. Such efforts might include the integration of β-tricalcium phosphate and other resorbable elements into the slurry system [18]. These types of composite

slurries would have base components with both rapid (β-tricalcium phosphate) and slower (HA) rates of resorption tailored toward supporting patient specific bone repair needs. Development of novel slurries for DLP printed materials for orthopedic and clinical applications would benefit from an in-depth material characterization analysis (i.e. FTIR spectrum, particle size distribution within slurries, Ca/P ratio and degree of carbonation sintered prints) to aid in apprehension of cell response to DLP produced ceramic components. Microarchitecture of DLP printed materials can also be designed to have enhanced osteoconductive properties [32]. Assessment of bone formation and vascularization in ectopic small animal models will guide the design of a resorbable scaffold for restoring native bone in response to traumatic bone defects.

The results reported here further advance the bone defect repair field toward the ultimate realization of a scalable fabrication system which can support the development of fully resorbable, biocompatible scaffolds. In addition, applying the DLP process to scaffold design will provide architectural freedom to build complex 3D structures engineered with the biomechanics representative of bone [50,61]. Overcoming these divergent constraints will enable patient specific long-term assimilation of the bone substitute with native tissue environments, restoring function to trauma-induced bone damage.

## Supporting information

**S1 Fig. Build orientation of coupons for biological testing.** (A) Geometry of the build was a circular disc with a hole on the edge to facilitate handling with sterile forceps. (B) The flat bottom is adhered to the build plate and the layers are built up in the direction of the arrow. (TIF)

**S2 Fig. Phase identification using X-ray diffraction, Fourier-transform Infrared Spectroscopy, and Inductively Coupled Plasma-Optical Emission Spectroscopy.** (A) The fired coupon was analyzed by X-ray diffraction (XRD) to evaluate the crystalline phases. The XRD pattern was consistent with hydroxyapatite and showed only three very minor secondary phase peaks, which were consistent with orthorhombic hydroxyapatite and magnesium oxide (arrows in inset). (B) The coupon was also analyzed by Fourier-transform Infrared Spectroscopy (FTIR), revealing the IR spectrum of LithaBone HA400 powder in diamond transmission cell. (C) Inductively Coupled Plasma-Optical Emission Spectroscopy (ICP-OES) was used to characterize the average calcium/phosphate (Ca/P) ratio of coupons. Analysis was done on fired (sintered) LithaBone HA400 powder (Sample IDs S22-01607a-d). Wt%, Mol%, and Ca/P ratio was measured. (TIF)

**S3 Fig. Characterization of surface roughness using multi-line measurement technique.** (A) Image of coupon at 12x magnification showing location of five 2 mm parallel lines in vertical and horizontal directions. Example multi-line roughness profiles in the (B) vertical and (C) horizontal directions with (D) values displayed in the table. (TIF)

**S4 Fig. Contact angle approach to measure hydrophilicity of coupons.** (A) Pictures of water droplet on coupon surface, image taken 2 seconds after liquid contacted the surface of the coupon, resulting in a contact angle of (L,R) 73.5˚, 74.5˚ and (B) a contact angle of (L,R) 0˚, 0˚ with the water fully spread across coupon surface. Images demonstrate the contact angle approach used to measure hydrophilicity of coupons. (TIF)

**S5 Fig. Extractable and leachables study using U2OS cells.** Seeded at ~66% confluency ($4.6x10^3$ cells/cm$^2$), U2OS cells were exposed to coupon conditioned culture media for 48 hours. Cells cultured on tissue culture polystyrene (TCP) and in the presence of coupon conditioned culture media were observed to maintain high confluency with no negative impact on morphology or proliferative capabilities. Image was taken using EVOS, phase contrast, 10x objective magnification, scale bar is 250μm.
(TIF)

**S6 Fig. Uncoated DLP printed coupons have reduced cell attachment efficiency.** (A) MSCs were seeded at a targeted density of ~25% ($6.25x10^3$ cells/cm$^2$) onto tissue culture polystyrene (12-well plates, "TCP") and coupons. Image analysis was used to measure total number of nuclei 24 hours after seeding for a surface area of ~3cm$^2$. Coupons have ~30% fewer number of cells compared to tissue culture polystyrene, "TCP". (B) Cell coverage was measured using image analysis (by image J) of uncoated HA coupons and coated coupons (fibronectin and Matrigel). Cell coverage was measured as the area covered by cells divided by total area (n = 4). (C) MSCs cultured for 35 days on HA coupons coated with fibronectin or Matrigel had greater confluency (red and pink stained region are cells, cultured in standard MSC culture media). Uncoated HA coupons had lower percent cell coverage compared to coated HA coupons (fibronectin or Matrigel, collagen). Coating HA coupons-maintained confluency of nearly 100% for up to the 35$^{th}$ day of culture. Cells were stained with nuclear fast red and have a pink/fuchsia color. Scale bar 1.25mm, Leica dissection microscope was used to image surfaces of HA coupons at 4x objective magnification (magnified images).
(TIF)

**S7 Fig. Cell count of U2OS cells using indirect and direct measurements.** (A) RealTime-Glo$^{TM}$ MT Cell Viability Assay was used to measure relative luminescence (RLUs) of U2OS cells seeded on standard tissue culture polystyrene (12-well plates) at cell densities to calculate a linear equation correlating RLUs with total cells. (B) To analyze accuracy of linear equation and determine functionality of the RealTime-Glo$^{TM}$ MT Cell Viability Assay, U2OS cells were seeded at different target densities starting ~25% (1x) and increasing by a factor of 2x and 3x. Total number of cells were indirectly measured (calculated based on RLU measurements) for incubation time points 4, 22, 29, and 48 hours and directly (using image analysis) for time point 66 hours.
(TIF)

**S8 Fig. Alizarin Red S staining of MSCs cultured for 14 days on standard polystyrene tissue culture plates.** (A-C) MSCs were cultured for 14 days in either osteogenic differentiation media or (D-F) non-differentiation media. On day 14, cells were stained with Alizarin Red S dye. (B and E) Images taken at 4x objective magnification and (C and F) at 20x objective magnification. (B and E) Scale bar is 1.25mm and (C and F) 0.25mm.
(TIF)

**S9 Fig. Von Kossa staining of MSCs cultured for 35 days on standard polystyrene tissue culture plates.** (A-C) MSCs were cultured for 35 days in either osteogenic differentiation media or (D-F) non-differentiation media. On day 35, cells were von Kossa stained. (B and E) Images taken at 4x objective magnification and (C and F) at 20x objective magnification. (B and E) Scale bar is 1.25mm and (C and F) 0.25mm.
(TIF)

**S10 Fig. Flexural testing and proposed DLP designs to investigate bone formation.** (A) Illustrations of an example build plate for DLP 3D printing of LithaBone HA400 horizontal

and vertical rupture test bars for a four-point bend flexural test set up, where flexural strength is measured for vertical (top) and horizontal (bottom) 3D printed bars. (B) Plot of flexural strength (MPa) versus open porosity (%) for fired (sintered) LithaBone HA400 modulus of rupture test bars, depend on their print orientation (horizontal vs vertical). (C) Clover and tri-furcating LithaBone HA400 scaffold designs for proposed use in bone formation experimental studies. Images of scaffold were taken at 200x magnification using a Hirox 3D digital microscope and right most image is of a clover design LithaBone HA400 scaffold stained with nuclear fast (pink/red; this stain is part of the von Kossa staining protocol and it stains nuclei) to visualize presence of U2OS cells after 14 days of culture. Scale bar is 1mm.
(TIF)

## Acknowledgments

We would like to thank the General Electric Company and colleagues at GE Research, particularly Kenneth Conway for consultation on best practices for sterilization and Liz McDonough in the Bioscience organization for her extensive help in using the Cell Dive imaging platform.

## Author Contributions

**Conceptualization:** Jessica S. Martinez, Sara Peterson, Cathleen A. Hoel, Daniel J. Erno, Tony Murray, Fiona Ginty, Steven J. Duclos, Brian M. Davis, Gautam Parthasarathy.

**Data curation:** Jessica S. Martinez, Sara Peterson, Cathleen A. Hoel, Daniel J. Erno, Tony Murray, Linda Boyd, Jae-Hyuk Her, Nathan Mclean, Robert Davis, Brian M. Davis, Gautam Parthasarathy.

**Formal analysis:** Jessica S. Martinez, Sara Peterson, Cathleen A. Hoel, Daniel J. Erno, Tony Murray, Linda Boyd, Jae-Hyuk Her, Nathan Mclean, Robert Davis, Fiona Ginty, Steven J. Duclos, Brian M. Davis, Gautam Parthasarathy.

**Funding acquisition:** Jessica S. Martinez, Sara Peterson, Cathleen A. Hoel, Daniel J. Erno, Tony Murray, Steven J. Duclos, Brian M. Davis, Gautam Parthasarathy.

**Investigation:** Jessica S. Martinez, Sara Peterson, Cathleen A. Hoel, Daniel J. Erno, Tony Murray, Nathan Mclean, Robert Davis, Brian M. Davis, Gautam Parthasarathy.

**Methodology:** Jessica S. Martinez, Sara Peterson, Cathleen A. Hoel, Daniel J. Erno, Tony Murray, Brian M. Davis, Gautam Parthasarathy.

**Project administration:** Jessica S. Martinez, Cathleen A. Hoel, Steven J. Duclos, Brian M. Davis, Gautam Parthasarathy.

**Supervision:** Fiona Ginty, Steven J. Duclos, Brian M. Davis, Gautam Parthasarathy.

**Validation:** Jessica S. Martinez, Sara Peterson, Cathleen A. Hoel, Daniel J. Erno, Tony Murray, Brian M. Davis, Gautam Parthasarathy.

**Visualization:** Jessica S. Martinez, Sara Peterson, Cathleen A. Hoel, Daniel J. Erno, Fiona Ginty, Steven J. Duclos, Brian M. Davis, Gautam Parthasarathy.

**Writing – original draft:** Jessica S. Martinez, Sara Peterson, Cathleen A. Hoel, Daniel J. Erno, Fiona Ginty, Steven J. Duclos, Brian M. Davis, Gautam Parthasarathy.

**Writing – review & editing:** Jessica S. Martinez, Sara Peterson, Cathleen A. Hoel, Daniel J. Erno, Fiona Ginty, Steven J. Duclos, Brian M. Davis, Gautam Parthasarathy.

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
