## [Decision Letter · Decision Letter 0]

13 Dec 2021

PONE-D-21-34494High resolution DLP stereolithography to fabricate biocompatible hydroxyapatite structures that support osteogenesis.PLOS ONE

Dear Dr. Martinez,

Thank you for submitting your manuscript to PLOS ONE. After careful consideration, we feel that it has merit but does not fully meet PLOS ONE’s publication criteria as it currently stands. Therefore, we invite you to submit a revised version of the manuscript that addresses the points raised during the review process.

We look forward to receiving your revised manuscript.

Kind regards,

Aldo Boccaccini

Academic Editor

PLOS ONE

Journal Requirements:

2.Thank you for stating the following financial disclosure: 

(This work was funded by a research grant from GE Research. The authors Jessica S. Martinez, Sara Peterson, Cathleen A. Hoel, Daniel J. Erno, Tony Murray, Linda Boyd, Jae-Hyuk Her, Fiona Ginty, Steven J. Duclos, Brian M. Davis, and Gautam Parthasarathy are employed by General Electric (GE), with specific affiliation at GE Research. All authors contributed to the design of the studies and Jessica S. Martinez, Sara Peterson, Cathleen A. Hoel, Daniel J. Erno, Steven J. Duclos, Brian M. Davis, and Gautam Parthasarathy co-wrote the manuscript. The funders had no role in study design, data collection and analysis, decision to publish, or preparation of the manuscript.)  

We note that one or more of the authors is affiliated with the funding organization, indicating the funder may have had some role in the design, data collection, analysis or preparation of your manuscript for publication; in other words, the funder played an indirect role through the participation of the co-authors. If the funding organization did not play a role in the study design, data collection and analysis, decision to publish, or preparation of the manuscript and only provided financial support in the form of authors' salaries and/or research materials, please do the following:

a. Review your statements relating to the author contributions, and ensure you have specifically and accurately indicated the role(s) that these authors had in your study. These amendments should be made in the online form.

b. Confirm in your cover letter that you agree with the following statement, and we will change the online submission form on your behalf: 

“The funder provided support in the form of salaries for authors [insert relevant initials], but did not have any additional role in the study design, data collection and analysis, decision to publish, or preparation of the manuscript. The specific roles of these authors are articulated in the ‘author contributions’ section.

(We would like to thank the General Electric Company, which funded this scientific work to advance basic scientific research in translational additive manufacturing efforts. We would also like to thank colleagues at GE Research, particularly Kenneth Conway for consultation on best practices for sterilization and Liz McDonough in the Bioscience organization for her extensive help in using the Cell Dive imaging platform.)

(This work was funded by a research grant from GE Research. The authors Jessica S. Martinez, Sara Peterson, Cathleen A. Hoel, Daniel J. Erno, Tony Murray, Linda Boyd, Jae-Hyuk Her, Fiona Ginty, Steven J. Duclos, Brian M. Davis, and Gautam Parthasarathy are employed by General Electric (GE), with specific affiliation at GE Research. All authors contributed to the design of the studies and Jessica S. Martinez, Sara Peterson, Cathleen A. Hoel, Daniel J. Erno, Steven J. Duclos, Brian M. Davis, and Gautam Parthasarathy co-wrote the manuscript. The funders had no role in study design, data collection and analysis, decision to publish, or preparation of the manuscript.)

Reviewers' comments:

Reviewer's Responses to Questions

**Comments to the Author**

1. Is the manuscript technically sound, and do the data support the conclusions?

Reviewer #1: Yes

Reviewer #2: Partly

2. Has the statistical analysis been performed appropriately and rigorously? 

Reviewer #1: Yes

Reviewer #2: I Don't Know

3. Have the authors made all data underlying the findings in their manuscript fully available?

Reviewer #1: Yes

Reviewer #2: Yes

4. Is the manuscript presented in an intelligible fashion and written in standard English?

Reviewer #1: Yes

Reviewer #2: Yes

5. Review Comments to the Author

Reviewer #1: The paper is in my view very well written and gives an excellent insight into the biological properties of the investigated 3D-printable hydroxapatite material. The structure of the text is appropriate and the presented results support the conclusions which the authors have drawn from their research. I have a minor and a major comment:

(1) In the section “Materials and methods” the author make no references to the figures, these references are made later in the section about “Results”. This sometimes makes it difficult to understand certain aspects in the text. I suggest to reference the relevant figures (e.g. sketch of working principle, geometry of coupons, …) also in the beginning of the text.

(2) I am missing some numbers on mechanical properties of the used materials, since the strength of the used HAP determines its usability in the clinical setup. Values on bending or biaxial bending strength would certainly add a lot of value to the paper.

Reviewer #2: This manuscript deals with hydroxyapatite scaffolds prepared via the DLP stereolithography technique and their interaction with two cell types. The general topic is relevant to the field of bone repair engineering and the production of calcium phosphate 3D scaffolds allowing for good cell behavior, especially in relation to their osteogenic character.

The paper is by itself rather well written. However, in my opinion, several improvements would enhanced the quality of the paper, especially aiming publication in a high IF journal like PlosOne.

One general criticism according to me is that the HA material itself is only very poorly detailed/characterized and the main results are focusing on cell response. Although cell response is indeed capital in this type of application, the characteristics of the biomaterial itself is also of prime importance. I have well noted that a commercial HA slurry was used, however a more detailed description of the HA characteristics would be quite necessary. At minimum, the XRD pattern, FTIR spectrum, particle size distribution, Ca/P ratio, eventual degree of carbonation etc... would be relevant to give to better apprehend the cell behavior results and their relevance to the field. “Minor impurities” are mentioned on page 12 but no more details are given. Again, which impurities, to which extent etc would help to strengthen the impact of the paper. The terms “photocurable HA slurry” are used but the curability is unclear. More details would be interesting so as to more clearly understand the type of interactions that may occur during processing (which resin? which mechanism of curing? …).

In the introduction section, comparisons are made with metal based systems. However, the relevance of metal-based protheses lies in their mechanical properties which is a point that does not seem to be considered, instead the authors mention the non resorbability of the metals, but this seems out of scope for metals. Please rethink this part.

Some terms do not seem adequate, like “plastics” instead of “polymers”, or the term “host”, both in page 4. Please check also some English terminology/spelling as for “biologic” versus “biological” on page 17.

On page 7 a heat treatment is mentioned at 1300°C. Again, this may be mirrored with the HA composition as nonstoichiometry and/or carbonation (or perhaps also the impurities that are mentioned once in the text) could lead to partial decomposition. This is also why better characterization/compositional details would be greatly appreciated.

To improve interactions with some cells, a “priming” with fibronectin or Matrigel is mentioned. But again, no chemistry data or quantification or proof of actual surface “functionalization” is shown to demonstrate the surface modifications undergone. Similarly, the “extractables” and “leachables” that are mentioned, e.g. on page 14, are only rapidly mentioned but deserve more attention. Which compounds are considered here? Did you run chemical analyses? Did you titrate chemically to determine the remaining amounts?

Concerning the osteosarcoma cell line, perhaps few details stating which this could be a good cell model for this study could be worth adding.

On page 17, the text right after the “Fig.5 title” seems to be the figure legend and not actual text.

On page 20, there is a mention of “TCP”. It is unclear which compound is considered here, please be more precise.

Finally, this 3D approach is notably pertinent if there is a need to make complex 3D shapes. It could have been a nice addition toward the end of the paper to show that this DLP approach can also lead to complex shapes… Also, some info about the limitations in the size of the 3D objects that can be prepared by this approach could have been interesting.

All in all, the paper deals with a relevant topic and shows some nice cell behavior with several staining and in vitro tests to demonstrate this. The weakness is more on the physico-chemistry which would gain to be developed into more details. This is why I propose re-submitting with “major” revisions in view of manuscript overall improvement.

6. PLOS authors have the option to publish the peer review history of their article (what does this mean?). If published, this will include your full peer review and any attached files.

Reviewer #1: No

Reviewer #2: No

---

## [Author Response · Author response to Decision Letter 0]

7 Jul 2022

We responses to all points of concerns addressed by the academic editor and reviewers in our response to reviewer word file outlining how each concern was addresses and highlighting where changes were made in the revised manuscript.

---

## [Decision Letter · Decision Letter 1]

18 Jul 2022

High resolution DLP stereolithography to fabricate biocompatible hydroxyapatite structures that support osteogenesis.

PONE-D-21-34494R1

Dear Dr. Martinez,

We’re pleased to inform you that your manuscript has been judged scientifically suitable for publication and will be formally accepted for publication once it meets all outstanding technical requirements.

Kind regards,

Aldo Boccaccini

Academic Editor

PLOS ONE

Additional Editor Comments (optional):

The authors have introduced satisfactorily the corrections requested by the referees.

Reviewers' comments:

Reviewer's Responses to Questions

**Comments to the Author**

1. If the authors have adequately addressed your comments raised in a previous round of review and you feel that this manuscript is now acceptable for publication, you may indicate that here to bypass the “Comments to the Author” section, enter your conflict of interest statement in the “Confidential to Editor” section, and submit your "Accept" recommendation.

Reviewer #1: All comments have been addressed

2. Is the manuscript technically sound, and do the data support the conclusions?

Reviewer #1: Yes

3. Has the statistical analysis been performed appropriately and rigorously? 

Reviewer #1: Yes

4. Have the authors made all data underlying the findings in their manuscript fully available?

Reviewer #1: Yes

5. Is the manuscript presented in an intelligible fashion and written in standard English?

Reviewer #1: Yes

6. Review Comments to the Author

Reviewer #1: This is an interesting paper. The required changes have been made and the paper can go into publication.

7. PLOS authors have the option to publish the peer review history of their article (what does this mean?). If published, this will include your full peer review and any attached files.

Reviewer #1: No

---

## [Editor Report · Acceptance letter]

29 Jul 2022

PONE-D-21-34494R1 

High resolution DLP stereolithography to fabricate biocompatible hydroxyapatite structures that support osteogenesis. 

Dear Dr. Martinez:

I'm pleased to inform you that your manuscript has been deemed suitable for publication in PLOS ONE. Congratulations! Your manuscript is now with our production department. 

Kind regards, 

on behalf of

Professor Aldo Boccaccini 

Academic Editor

PLOS ONE